# Testing of a novel questionnaire of Household Exposure to Wood Smoke

**Paula M. Meek**[1]*, **Sarah J. Schmiege**[2], **Akshay Sood**[3], **Hans Petersen**[4], **Rodrigo Vazquez-Guillamet**[5], **Hammad Irshad**[4], **Jacob McDonald**[4], **Yohannes Tesfaigzi**[6]

1 College of Nursing, University of Utah, Salt Lake City, Utah, United States of America, 2 University of Colorado Anschutz Medical Campus, Aurora, Colorado, United States of America, 3 Department of Medicine, School of Medicine, University of New Mexico, Albuquerque, New Mexico, United States of America, 4 Lovelace Respiratory Research Institute, Albuquerque, New Mexico, United States of America, 5 John T. Milliken Department of Medicine, Washington University School of Medicine, St. Louis, Missouri, United States of America, 6 Department of Medicine, Pulmonary and Critical Care Medicine, Brigham and Women's Hospital, Harvard Medical School, Boston, Mass, United States of America

* Paula.Meek@nurs.utah.edu

**Data Availability Statement:** De-identified data related to this manuscript are publicly available through Harvard Dataverse: https://doi.org/10.7910/DVN/B0OGCE.

## Abstract

Household air pollution from wood smoke (WS), contributes to adverse health effects in both low- and high-income countries. However, measurement of WS exposure has been limited to expensive in-home monitoring and lengthy face-to-face interviews. This paper reports on the development and testing of a novel, self-report nine-item measure of WS exposure, called the Household Exposure to Wood Smoke (HEWS). A sample of 149 individuals using household wood stoves for heating from western states in the U.S., completed the HEWS during the winter months (November to March) of 2013 through 2016 with 30 subjects having in-home particle monitoring. Hard copy or online surveys were completed. Cronbach's alpha (α), intraclass correlations (ICC), exploratory factor analysis (EFA) and tests of associations were done to evaluate reliability and validity of the HEWS. Based on initial analysis, only 9 of the 12 items were retained and entered in the EFA. The EFA did not support a unitary scale as the 9 items demonstrated a 3-factor solution (WS exposure *duration*, *proximity*, and *intensity*) with Cronbach's α of 0.79, 0.91, and 0.62, respectively. ICC was 0.86 of the combined items with single items ranging from 0.46 to 0.95. WS *intensity* was associated with symptoms and levoglucosan levels, while WS *duration* was associated with stove and flume maintenance. The three-dimensional HEWS demonstrated internal consistency and test-retest reliability, structural validity, and initial criterion and construct validity.

## Introduction

The purpose of this article was to report on the psychometric testing of a new self-report questionnaire developed to evaluate characteristics of household air during exposure to wood smoke (WS) and symptoms as a result of exposure.

Studies in low- and high-income countries have linked WS exposure with substantial health effects such as: increased prevalence of respiratory symptoms and chronic

**Funding:** This research was supported by grants from the NIH awarded to PM (award number: R15 HL115544). The funders had no role in study design, data collection and analysis, decision to publish, or preparation of the manuscript.

**Competing interests:** All authors have declared that no competing interests exist.

bronchitis; decreased pulmonary function; and increased risk of lung cancer from biomass smoke [1–6]. Although WS exposure has been investigated less often in high-income countries, it is estimated that 8.8 million homes use wood stoves in the United States [7,8]. The potential impact associated with WS exposure has become more recognized as a health risk [9–11].

Indoor WS exposure in the United States is prevalent, yet, there is substantial/major denial about the adverse health effects of WS [12]. The Environmental Protection Agency (EPA) confirms that WS is more dangerous than an equal amount of cigarette smoke, as WS contains 12 times more carcinogenic components and lasts 40 times longer in the body [6]. Despite efforts to remove or improve wood stoves over the past decades, wood stoves remain an essential source of heating homes, for many households [13].

The Lovelace Smokers' Cohort (LSC) is a well-characterized cohort of smokers living in the Albuquerque, NM area, (an urban high-altitude community). Studies of the LSC, have found that exposure to WS, increased the odds of both chronic airflow obstruction and chronic bronchitis by 56% as compared to those not similarly exposed [14]. These findings are consistent with studies in low-income countries [15,16]. Also in these studies, symptoms such as cough (productive and nonproductive), shortness of breath and in some cases wheezing, were associated with WS exposure [7,8]. We found an additive interaction between WS and current cigarette smoke exposure and the odds for COPD in the LSC [14]. Estimates are that close to 13% of Americans have smoked at least 100 cigarettes in their lifetime, placing this group at increased risk for WS-associated COPD [17]. Given these demographics, WS is an underappreciated, but important risk factor for respiratory disease in a substantial portion of Americans and individuals in high-income countries.

Understanding the augmented susceptibility, given combined exposures, is the next step in the prevention, early diagnosis, and management of chronic respiratory conditions. However, there are no clear means of measuring WS exposure, other than expensive in-home monitoring and lengthy face-to-face interviews. The World Health Organization (WHO) historically used an extensive survey, administered face-to-face, to report on WS exposure, burning conditions and symptoms during cooking and household work in low-income countries [18]. WS exposure in high-income countries differs in magnitude, type and pattern from low-income countries, as demonstrated by a recent report from Norway. They used a survey to determine the use of wood stoves and other combustible agents and their relationship to particle measurements, such as particles less than 2.5-μm aerodynamic diameter or PM2.5 [19]. However, there is still no psychometrically tested self-report measure of wood smoke, which could simplify screening and evaluating for WS exposure in high-income countries. Consequently, we modified the WHO survey questions to develop a novel, self-report measure of WS exposure, called the Household Exposure to Wood Smoke (HEWS), for use in high-income countries. This report presents the first testing of the HEWS.

## Methods

### Ethics statement

Human subject approval was obtained through the Colorado Multiple Institutional Review Board (#13–1470) Denver, CO and the Liberty Review Board, DeLand Florida (#13.12.0007). Formal consent was obtained in writing from those participants who agreed to in-home monitoring at the beginning of the home visit. Participants recruited through advertisement that virtually completed surveys formal consent was obtained in writing via REDcap or if they preferred verbally over the telephone.

## Study population

The sample for this investigation was recruited in two ways; first, they were drawn from those currently enrolled in the LSC who reported exposure to WS in the home. Further details of the LSC have been described previously [20]. The second sample was through recruitment across the western mountain states via radio, TV, newspaper and local flyers. Existing LSC participants were re-contacted to take part in the study during the heating season (November through February) and made up the enriched in-home subsample. We contacted 200 current LSC individuals and over 300 others from western mountain states. Initial screening occurred via telephone to determine eligibility. Individuals that responded to recruitment efforts were considered eligible to participate if they cooked or heated their homes with a wood stove for a period of at least 3 months of the year and were currently burning wood in their home daily during the sampling period.

## Study design

Sample size estimates were calculated based on the established standard of a minimum of 10 subjects per questionnaire item to be tested [21]. The 12 item HEWS would therefore require a minimum of 120 individuals. A final sample size of 149 was obtained. The research design was a cross-sectional sample, with repeat measurements in two subsamples. The first subsample (n = 30) was obtained to evaluate test retest reliability and criterion reference validity, by examining the in-home particle and levoglucosan concentrations [22,23]. This subsample was an enriched subsample, as it was obtained from the LSC, with a greater percentage of potential participants with chronic conditions and history of smoking. Subjects in this subsample completed the in-home paper and pencil version of the HEWS. The second subsample of participants (n = 61) were recruited from the western mountain states. This subsample also took the questionnaire one week apart but on-line, to determine the reliability test-retest ICC assessment of an online format. Additionally, in the in-home subsample, the relationship of the HEWS with in-home particle concentrations were measured. The two subsamples were used to determine the stability of responses. This study followed the COnsensus-based Standards for the selection of health status Measurement INstruments (COS- MIN) criteria for studies that investigate the psychometric properties of self-report measures [24–27].

## Development of HEWS

The HEWS questions were selected from the World Health Organization D7 household interview (as a reference) that is used to measure exposure to biomass fuels in low-income countries [18]. The HEWS version tested in this study, is a self-report survey that consists of 12 questions on a 1 to 4 scale, designed to elicit responses regarding WS exposure in three potential areas *duration*, *proximity* and *intensity*. These three areas potentially heighten the degree of exposure to inhaled smoke when wood is burned in the household. Initial face validity was examined by asking several individuals who participate in residential wood burning, and experts in questionnaire development, to examine the HEWS questions. All questions (items) were determined to be appropriate, with slight modifications to wording but not the 1 to 4 scoring or associated label.

To further determine the face validity of HEWS, five individuals from the LSC who reported exposure to WS, were contacted in a pilot study. The HEWS was administered to individuals, along with measurement of particle exposure in their home. All individuals completed the questions without difficulty. However, based on the responses to the questions, scaling was slightly modified on three items concerning exposure time. The pilot particulate

**Table 1. Index of exposure to household air wood smoke questions, item scale wording and potential exposure emphasis.**

| Items | Item Scale Wording | Exposure Emphasis |
|---|---|---|
| 1. During the past week, how many hours was wood burned in the house over 24 hours? | 1–3 days to Daily-all day | Duration |
| 2. During the last week, how often did you burn wood in your house? | 1–6 hour to 18 or more hours | Duration |
| 3. Over the past week when wood was burning in the stove/ fireplace, I could smell smoke in the house? | Never to Always | Intensity |
| 4. Over the past week was wood burning in the stove/ fireplace while you sleep? | Never to Always | Duration |
| 5. When wood is burning it is your job to look after the stove/ fireplace? | Never to Always | Proximity |
| 6. Over the past week when wood was burning in the stove/fireplace, there was some smoke in the room? | Never to Always | Intensity |
| 7. When wood is burning how close to the stove/ fireplace are you? | Greater than 6 feet to Under 1 foot | Proximity |
| 8. Usually, when wood was burning in the stove/ fireplace, I was in the same room? | Never to Always | Proximity |
| 9. Over the past week when you had wood burning in the stove/ fireplace the door/front of the stove / fireplace was open? | Never to Always | Intensity |
| 10. Over the past week when you had wood burning in the stove/ fireplace were the windows open? | Never to Always | Intensity |
| 11. On average over the past week how many hours were you in the room where wood was burning? | 1–2 hours to More than 10 hours | Duration |
| 12. Typically, it is your job to start the wood fire in the stove/fireplace? | Never to Always | Proximity |

samples collected from homes were extracted and analyzed by mass spectrometry for levoglucosan levels, a WS combustion product. In this pilot sample, levoglucosan concentrations trended with particulate matter concentrations, verifying that exposure to indoor WS occurred. Examination of the HEWS total score, and the indoor levels of particles and levoglucosan, showed a trend towards a positive relationship between the HEWS scores, and objective indoor measurements. These findings supported the need for further testing of the psychometric properties of the HEWS.

The final 12 questions of the HEWS tested in this investigation, are listed along with the wording for item scaling and potential exposure emphasis in Table 1. The HEWS was proposed to be unidimensional, with a total score with higher total score reflecting greater WS exposure.

## Study measures

Besides the HEWS, demographic information such as age, smoking history, and history of respiratory disease was obtained using the American Thoracic Society (ATS)-DLD-78 questionnaire, with some questions added about in-home exposures to smoking and animals. The St. George's Respiratory Questionnaire (SGRQ) was also used to evaluate respiratory health status, with higher scores indicating worse health status [28]. The SGRQ consists of a total score and scores on three subscales; impact of disease, activity limitation, and symptoms, with a difference of 4 points on the total score indicating a clinically meaningful difference [29]. A report of baseline symptoms experienced was obtained, including; chest congestion, cough, phlegm, wheeze, chest discomfort, shortness of breath, and tiredness recorded on a 1 to 5 scale (1 = not at all, 5 = extremely).

### In-home particulate samples and analysis

Particulate samples in the in-home subsample were obtained to determine the level of PM2.5 and levoglucosan concentrations. Levoglucosan (1, 6-anhydro-b-D-glucopyranose), a cellulose combustion product, is a tracer species for WS, mainly because of its high resistance to degradation [30,31]. Samples were obtained over a seven day period, using an in-home particulate monitor placed in a common living area of the home during the winter months. Filter samples were collected with a Personal Environmental Monitor (PEM, Model 200, PEM-10-2.5, MSP Corporation, Shoreview, MN). Filter samples were analyzed gravimetrically to calculate PM2.5 concentrations, and chemically to determine the concentrations of levoglucosan. However, due to storage and chemical analysis failure, levoglucosan values were obtained in only 23 of 30 homes. The levoglucosan values showed a linear association with PM2.5 ($R^2 = 0.83$) concentrations, supporting the linkage between the particle concentration and WS. Details of the methods used to collect the particle samples and chemically extract the levoglucosan are previously published [32].

### Statistical analysis

A comparison of the three subsamples (in-home and online test-retest and baseline only) was conducted to look for sample differences. Summary statistics for continuous variables consisted of means and standard deviations (S.D.), analyzed using one-way analysis of variance. Categorical variables are presented as proportions and analyzed using Chi-square tests.

All analyses were conducted using SAS version 9.4 (Cary, NC). Instrument analysis was guided by the consensus-based standards for the selection of health measurement instruments (COSMIN) and began with analysis [33] to answer three key questions: 1) Do the HEWS items have a potential distribution of scores that will impede detection of differences in WS exposure (floor or ceiling effects)? 2) Do the HEWS items demonstrate an adequate item-to-item correlation (.30-.70)? 3) Are the HEWS item ICC values (>.60) adequate in terms of test-retest reliability (*i.e.* stability) [23].

Exploratory factor analysis (EFA) was used to evaluate the proposed structure of the HEWS and to evaluate for unidimensionality. A principal axis factor extraction method was used with an oblique (Promax) rotation. The EFA model results were examined for the overall fit of the proposed unidimensional measurement model and the magnitude of the factor loadings of each item, with a cutoff of 0.40. Although a unidimensional structure was initially hypothesized, models with 1 to 4 factors were evaluated. The final model was selected based on model convergence, model interpretability, and the observed scree plot/eigenvalues. Once the final factor structure was chosen, the internal consistency reliability of the items loading on each factor was examined using Cronbach's alpha. Composite subscale scores were then created by calculating the mean of the items in a given factor.

Following factor extraction, we conducted Pearson's correlations to examine associations of the HEWS with the total and SGRQ subscale scores (*i.e.* impact, activity, and symptoms), baseline symptoms, and in-home particulate measures (n = 30). Independent samples t-tests were used to assess mean factor scores based on whether or not the stove was maintained regularly (yes/no) or the flue cleaned (yes/no).

## Results

### Demographic characteristics of subjects

The final sample consisted of 149 individuals. Complete data (with two test-retest) were obtained for the in-home (n = 30) and online (n = 61) subsamples. The remainder only

**Table 2. Demographic characteristics of three enriched subsamples comprising the final sample of n = 149.**

| | Test-retest Subsamples | | Baseline only n = 58 |
|---|---|---|---|
| | In-home n = 30 | Online n = 61 | |
| | M(sd) or n(%) | M(sd) or n(%) | M(sd) or n(%) |
| **Age** (% 65 years or older) | 11(36.7) | 11(18.0) | 13 (22.4) |
| **Gender** (% Female) | 17(56.7) | #37 (62.7) | §35 (62.5) |
| **Hispanic** (%) | *16(53.3) | 10(16.4) | §6 (10.7) |
| **Race (%)** | | | |
| Caucasian | 29(96.7) | ‡59 (96.7) | §54(93.1) |
| Other | 1(3.3) | 1(1.6) | 1(1.7) |
| **Reported Chronic Illness** | *20(66.7) | 18(29.5) | 18(31.0) |
| **Currently Smoking** | *8(26.7) | 4(6.6) | †9(15.8) |
| **Education** (% <high school) | N/A | 0 | §3(5.4) |
| **Income** (% did not have sufficient income to meet needs) | N/A | 1(1.6) | §4(7.1) |
| **Stove maintained in last year** (%Yes) | *18/29(62.1) | 59(95.7) | 53/57(93.0) |
| **Flue cleaned in the last year** (%Yes) | *15(50.0) | 50(82.0) | 52/57(91.2) |
| **Used a humidifier**(%Yes) | 9(30.0) | 20(32.8) | 22(37.9) |
| **Used an air filter** (%Yes) | 1(3.3) | 12(19.7) | 12(20.7) |
| **SGRQ Symptoms** | 23.4(18.9) | 18.5 (21.8) | 16.8(20.8) |
| **SGRQ Activity** | 23.7(17.9) | 16.0(23.2) | 13.7(21.8) |
| **SGRQ Impact** | 10.3(16.6) | 4.9(13.4) | 5.0(12.6) |
| **Baseline symptom presence** | | | |
| Did your chest feel congested today? | 0.29(0.5) | 0.31(0.7) | 0.26(0.6) |
| How often did you cough today? | 0.60(0.7) | 0.59(0.8) | 0.50(0.8) |
| Did you cough up any mucous (phlegm)? | 0.37(0.6) | 0.41(0.7) | 0.22(0.5) |
| Did you have chest discomfort today? | 0.14(0.4) | 0.20(0.5) | 0.10(0.4) |
| Did you feel short of breath today? | 0.18(0.4) | 0.31(0.7) | 0.14(0.4) |
| Did you feel tired or weak today? | 0.21(0.4) | 0.39(0.7) | 0.40(0.6) |

*Chi square or One way analysis significance < .05

# = 59 subjects responded to the question

§ = 56 subjects responded to the question

† = 59 subjects responded to the question

‡ = 60 subjects responded to the question, N/A: Data not available.

completed baseline measures (n = 58). The in-home subsample differed demographically from the other two samples with respect to ethnicity, history of chronic illness, current smoking, and stove and flue maintenance (Table 2). In the online test-retest subsample, most subjects were Caucasian women with fewer current smokers and lower prevalence of self-reported chronic illnesses. In addition, fewer subjects in the in-home test-retest subsample reported maintaining their stove or cleaning the flue in the last year.

## Item analysis

Initial examination of the items for means and standard deviations eliminated only one HEWS item, # 10 that asked about whether a window was left open during wood-burning. Elimination was due to an extreme floor effect, as essentially no one left their window open, likely due to cold weather. Several items had very low means (Table 3), and inadequate item to item correlations (Table 4).

**Table 3. Item analysis: Item means, and Interclass correlations by item.**

| Items | N = 149 Mean (SD) | Total n = 91 | Test-retest Subsamples ICC (CI) In-home n = 30 | Online n = 61 |
|---|---|---|---|---|
| 1. During the past week, how many hours was wood burned in the house over 24 hours? | 2.75 (1.06) | .77 (.52-.89) | .81 (.59-.91) | .62 (.37-.77) |
| 2. During the last week, how often did you burn wood in your house? | 2.70 (1.20) | .49 (-.07-.76) | .77 (.52-.89) | .70 (.50-.82) |
| 3. Over the past week when wood was burning in the stove/ fireplace I could smell smoke in the house? | 0.74 (0.64) | .77 (.51-.89) | .49 (-.07-.76) | .74 (.56-.84) |
| 4. Over the past week was wood burning in the stove/ fireplace while you sleep? | 1.86 (1.11) | .91 (.82-.96) | .77 (.51-.89) | .84 (.73-.90) |
| 5. When wood is burning, it is your job to look after the stove/ fireplace? | 1.99 (0.93) | .62 (.21-.82) | .91 (.82-.96) | .87 (.79-.92) |
| 6. Over the past week when wood was burning in the stove/fireplace there was some smoke in the room? | 0.69 (0.61) | .73 (.43-.87) | .62 (.21-.82) | .78 (.64-.87) |
| 7. When wood is burning how close to the stove/ fireplace are you? | 1.44 (0.63) | .80 (.57-.90) | .73 (.43-.87) | .46 (.11-.68) |
| 8. Usually when wood was burning in the stove/ fireplace I was in the same room? | 1.91 (0.68) | .95 (.89-.98) | .80 (.57-.90) | .81 (.68-.88) |
| 9. Over the past week when you had wood burning in the stove/ fireplace the door/front of the stove / fireplace was open? | 0.63 (0.81) | .61 (.18-.82) | .95 (.89-.98) | .89 (.81-.93) |
| 11. On average over the past week how many hours were you in the room where wood was burning? | 2.89 (1.05) | .87 (.73-.94) | .61 (.18-.82) | .67 (.46-.81) |
| 12. Typically, it is your job to start the wood fire in the stove/fireplace? | 1.97 (0.95) | .82 (.71-.90) | .87 (.73-.94) | .95 (.92-.97) |

SD = standard deviation, ICC = interclass correlations by time, CI = 95% confidence interval.

**Table 4. HEWS item to total and item to item correlations.**

| | Total | Q1 | Q2 | Q3 | Q4 | Q5 | Q6 | Q7 | Q8 | Q9 | Q11 | Q12 |
|---|---|---|---|---|---|---|---|---|---|---|---|---|
| +Q1 | 0.46 | 1 | | | | | | | | | | |
| +Q2 | 0.49 | ****0.63 | 1 | | | | | | | | | |
| Q3 | 0.24 | -0.02 | 0.13 | 1 | | | | | | | | |
| Q4 | 0.49 | ****0.61 | ***0.59 | 0.08 | 1 | | | | | | | |
| Q5 | 0.33 | 0.09 | 0.11 | 0.04 | 0.13 | 1 | | | | | | |
| Q6 | 0.29 | 0.10 | 0.12 | ****0.56 | *0.20 | 0.13 | 1 | | | | | |
| +Q7 | -0.03 | -0.09 | -0.07 | 0.07 | **-0.21 | -0.02 | 0.02 | 1 | | | | |
| Q8 | 0.22 | 0.08 | 0.04 | 0.01 | 0.01 | 0.06 | 0.02 | ***0.28 | 1 | | | |
| Q9 | 0.19 | 0.07 | 0.01 | *0.18 | 0.04 | 0.04 | ***0.30 | 0.002 | 0.18 | 1 | | |
| +Q11 | 0.52 | ****0.35 | ****0.37 | *0.20 | ****0.34 | 0.14 | 0.11 | 0.07 | ****0.42 | **0.24 | 1 | |
| Q12 | 0.30 | 0.02 | 0.10 | 0.09 | 0.14 | ****0.84 | 0.03 | -0.04 | 0.04 | 0.01 | *0.16 | 1 |

+ = items with alternate scaling wording

* $\leq$ 0.05

** $\leq$ 0.01

*** $\leq$ 0.001

**** $\leq$ 0.0001.

The item to item correlations ranged from r = -0.21 (between items 4 and 7) to r = 0.84 (between items 5 and 12). The item correlations were carefully reviewed for a pattern related to item scale wording or a method influence related to items 1, 2, 7 and 11 (Table 1), which was not seen. If item scale wording impacted responses, the correlations among items 1, 2, 7, and 11 would be expected to be stronger than those variables' correlations to other items. Item 7 in particular, was not well correlated with other items; in addition to the aforementioned *negative* correlation with item 4, the only other significant association was a small correlation with item 8 (r = 0.28, p<0.001). As another illustration, items 1, 2, and 11 were significantly correlated with item 4 (r = 0.61 for item 1, r = 0.59 for item 2, and r = 0.34 for item 11), despite not sharing the same scale wording; in contrast, item 4 had nonsignificant correlations with all other similarly scaled items, except for a small correlation with item 6 (r = 0.20). Reliability in terms of stability was assessed in the subsamples, using ICC calculation to determine consistency in response to the items. The majority of items were considered stable in both groups, but more so with the two subgroups together (Table 3). The exception to this was item 2, which had a low ICC (0.49) with the total sample, but strong ICCs in the separate subsample of in-home and online subsamples (0.77 and 0.70, respectively).

## Structural validity—exploratory factor analysis

The results of the EFA with principal axis factoring did not support a unidimensional tool; three factors were instead identified all of which had eigenvalues over 1 (Table 5). Factor one, labeled *duration*, consisted of four items (1, 2, 4 and 11) that asked about the quantity of wood burning in the home and represents the WS period of exposure. Factor two, labeled *proximity*, was composed of two items (5 and 12) that asked about the individual's responsibilities to start

**Table 5. Rotated factor loadings and eigenvalues obtained from three factor exploratory factor analysis (EFA) using principal axis factoring with promax rotation.**

| | Exposure | | |
|---|---|---|---|
| | Duration | Proximity | Intensity |
| Items | Factor 1 (2.29) | Factor 2 (1.42) | Factor 3 (1.06) |
| 1. During the past week, how many hours was wood burned in the house over 24 hours? | 0.770 | -0.002 | 0.028 |
| 2. During the last week, how often did you burn wood in your house? | 0.740 | 0.042 | 0.098 |
| 3. Over the past week when wood was burning in the stove/ fireplace I could smell smoke in the house? | 0.014 | 0.020 | 0.639 |
| 4. Over the past week was wood burning in the stove/ fireplace while you sleep? | 0.747 | 0.083 | 0.091 |
| 5. When wood is burning, it is your job to look after the stove/ fireplace? | 0.082 | 0.877 | 0.077 |
| 6. Over the past week when wood was burning in the stove/fireplace there was some smoke in the room? | 0.082 | 0.022 | 0.656 |
| 7. When wood is burning how close to the stove/ fireplace are you? | -0.162 | -0.020 | 0.172 |
| 8. Usually when wood was burning in the stove/ fireplace I was in the same room? | 0.068 | 0.056 | 0.265 |
| 9. Over the past week when you had wood burning in the stove/ fireplace the door/front of the stove / fireplace was open? | 0.040 | 0.013 | 0.400 |
| 11. On average over the past week how many hours were you in the room where wood was burning? | 0.434 | 0.121 | 0.361 |
| 12. Typically, it is your job to start the wood fire in the stove/fireplace? | 0.062 | 0.886 | 0.051 |

the fire or for maintaining the stove. These items represented the distance the individual was from WS exposure. Factor three, labeled *intensity*, consisted of 3 items (3, 6 and 9) that asked about the presence of smoke and whether the stove door was open during wood-burning. These items represented the potential for greater WS exposure. There were two items (7 and 8) that did not load adequately on any factor and asked about the individual's location in relation to the stove. Items 7 and 8 were therefore dropped from further analysis, resulting in a 9-item final questionnaire.

For the total tool, Cronbach's α was 0.70, with the three WS exposure factors of duration, proximity, and intensity 0.79, 0.91, and 0.62 respectively. The alphas were considered acceptable for a new tool. The three factor ICC's were 0.68, 0.72 and 0.66 (respectively), which were also considered acceptable.

## Additional validity

The intensity factor was correlated with the SGRQ and all baseline symptoms (Table 6). The duration factor was also correlated with subjects' reports of being tired. No significant correlations were found between any of the factor scores and the presence of particles, measured as FS concentration in the in-home subsample. However, despite a small sample (n = 23 with complete data), levoglucosan values correlated significantly with proximity (r = 0.43, p = 0.04) and intensity (r = 0.49, p = 0.02).

Mean scores on each of the three factors were examined in terms of whether participants consistently maintained the stove or cleaned the flue. Duration score was significantly higher (mean difference of 0.87, p < 0.001) among those who did not consistently maintain their stove, while proximity and intensity scores did not differ (mean differences were 0.0009 and 0.015, respectively), suggesting that those who cleaned their stove regularly actually used it less. Intensity was 0.26 points higher among those who did not clean the flue (p = 0.03) whereas duration (mean difference 0.19, p = 0.28) and exposure distance (mean difference 0.21, p = 0.21) did not differ depending on whether the flue was cleaned. These results should be interpreted with caution given the small number of participants who reported not consistently maintaining the stove (n = 17) or cleaning the flue (n = 31).

**Table 6. Correlations with SGRQ, baseline daily symptom report, and in-home measures of Levoglucosan.**

|  | Exposure | | |
|---|---|---|---|
|  | **Duration** | **Proximity** | **Intensity** |
| SGRQ–Symptoms | 0.05 | 0.04 | ***0.41 |
| SGRQ–Activity | -0.02 | -0.02 | ***0.32 |
| SGRQ—Impact | 0.03 | 0.07 | **0.25 |
| Did your chest feel congested today? | 0.003 | 0.044 | ***0.314 |
| How often did you cough today? | 0.062 | 0.042 | ***0.299 |
| Did you cough up any mucous (phlegm)? | 0.083 | 0.044 | **0.228 |
| Did you have chest discomfort today? | 0.003 | 0.019 | **0.243 |
| Did you feel short of breath today? | 0.065 | -0.042 | **0.263 |
| Did you feel tired or weak today? | *0.178 | -0.028 | **0.244 |
| Levoglucosan (n = 23) | 0.30 | *0.43 | *0.49 |

* <0.05

** <0.01

***<0.001.

## Discussion

The HEWS is the first psychometrically tested, self-report measure to examine wood burning patterns and exposure in a high-income country. The results show that the HEWS is not unidimensional but represents three distinct key aspects of exposure to WS, i.e., duration (factor 1), proximity (factor 2), and intensity (factor 3). Test-retest reliability was assessed in two subsamples (in-home and online) for a total of 91 individuals, with the results supporting stability. Preliminary evidence of internal consistency was seen for the HEWS dimensions. Further analysis may be needed with a larger sample to confirm structural validity. We presented evidence of preliminary criterion and construct validity that supported the HEWS dimensions of intensity correlating with levoglucosan values and reported symptoms. In addition, the HEWS dimension of proximity also demonstrated initial support for criterion validity given the correlation with levoglucosan. However, these findings must be replicated in a larger sample due to the wide variability in the particle concentration and levoglucosan measures found in the homes sampled. Finally, there is initial support for discriminant validity with the duration scores, relative to whether the individual stoves and flume had been cleaned, with the intensity score also discriminating among those reports of cleaning the flue. We have demonstrated that the HEWS is comprehensible, easily completed both in paper and online, internally consistent, stable and valid in a heterogeneous sample of individuals.

Further the HEWS was clearly associated with several important symptoms such as cough, phlegm, shortness of breath, chest discomfort, congestion, as well as fatigue with the intensity of WS. This was also found in relation to the SGRQ subscales of symptoms, activity and impact. The proximity to the wood burning did not demonstrate a relationship to symptoms in this sample. However, the duration of wood burning did have a small, but significant relationship to feeling tired or weak. Previous investigations in low-income countries have had limited, but significant reports of symptoms such as cough and phlegm, but these findings are uniquely associated with the intensity of the WS and not its mere presence. In addition, the majority of studies of WS in low-income countries, have not reported on specific symptom associations other than cough or wheeze in children [34,35].

To our knowledge, the HEWS represents a novel self-report questionnaire that can help identify individuals in high-income countries that may be at risk for exposure to WS. Previous investigations of self-reports of wood burning patterns and exposures have found a relationship between these reports and PM2.5, but did not assess levoglucosan to determine if there was direct relationship with WS, versus other combustible sources, such as candles or cigarette smoke [19]. However this study is not without limitations. This study used a convenience sample and excluded those who did not speak and understand English, and those who were unwilling to participate in an exploratory study in which there was no expectation of immediate individual benefit. In addition, there were some limitations due to cost and logistics of in-home sampling that prevented more home samples for particle and chemical analysis being obtained. Currently, the HEWS is being used with a larger in-home particle measurement sample and additional analysis such as confirmatory factor analysis and further exploration of potential methods variance.

The development and testing of the HEWS represents the first attempt to rigorously test a self-report measure of WS exposure for ongoing use in high-income countries. While there is clear evidence in low-income countries of the health hazards associated with biomass fuel and wood-burning, there is a need for a reliable, valid and simple to use questionnaire, such as the HEWS to better assess the magnitude of exposure of this hazard in high-income countries. Although more testing of the HEWS is indicated and ongoing, these initial results, in a relatively large and heterogeneous sample, suggest that the questionnaire shows promise of being useful for assessment of an individual's exposure to WS and their associated symptoms.

## Author Contributions

**Conceptualization:** Paula M. Meek, Jacob McDonald, Yohannes Tesfaigzi.

**Data curation:** Paula M. Meek, Hammad Irshad, Jacob McDonald, Yohannes Tesfaigzi.

**Formal analysis:** Paula M. Meek, Sarah J. Schmiege, Akshay Sood, Hans Petersen, Hammad Irshad, Jacob McDonald, Yohannes Tesfaigzi.

**Funding acquisition:** Paula M. Meek, Sarah J. Schmiege, Akshay Sood, Hans Petersen, Yohannes Tesfaigzi.

**Investigation:** Paula M. Meek, Akshay Sood, Hans Petersen, Rodrigo Vazquez-Guillamet, Yohannes Tesfaigzi.

**Methodology:** Paula M. Meek, Sarah J. Schmiege, Akshay Sood, Hans Petersen, Jacob McDonald, Yohannes Tesfaigzi.

**Project administration:** Paula M. Meek, Yohannes Tesfaigzi.

**Resources:** Paula M. Meek, Akshay Sood, Yohannes Tesfaigzi.

**Software:** Paula M. Meek, Sarah J. Schmiege, Hans Petersen, Hammad Irshad.

**Supervision:** Paula M. Meek, Akshay Sood, Yohannes Tesfaigzi.

**Validation:** Paula M. Meek, Akshay Sood, Yohannes Tesfaigzi.

**Visualization:** Paula M. Meek.

**Writing – original draft:** Paula M. Meek, Sarah J. Schmiege, Akshay Sood, Hans Petersen, Rodrigo Vazquez-Guillamet, Jacob McDonald, Yohannes Tesfaigzi.

**Writing – review & editing:** Paula M. Meek, Sarah J. Schmiege, Akshay Sood, Hans Petersen, Rodrigo Vazquez-Guillamet, Hammad Irshad, Jacob McDonald, Yohannes Tesfaigzi.

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
