## [Decision Letter · Decision Letter 0]

17 Oct 2022

PGPH-D-22-01097

Testing of a Novel Questionnaire of Household Exposure to Wood Smoke

Dear Dr. Meek,

Thank you for submitting your manuscript to PLOS Global Public Health. After careful consideration, we feel that it has merit but does not fully meet PLOS Global Public Health’s publication criteria as it currently stands. Therefore, we invite you to submit a revised version of the manuscript that addresses the points raised during the review process.

We look forward to receiving your revised manuscript.

Kind regards,

Changwoo Han, M.D., Ph.D.

Academic Editor

Journal Requirements:

2. Please send a completed 'Competing Interests' statement, including any COIs declared by your co-authors. If you have no competing interests to declare, please state "The authors have declared that no competing interests exist". Otherwise please declare all competing interests beginning with the statement "I have read the journal's policy and the authors of this manuscript have the following competing interests:"

3. Please amend your detailed Financial Disclosure statement. This is published with the article. It must therefore be completed in full sentences and contain the exact wording you wish to be published.

Additional Editor Comments (if provided):

Thank you for sharing very interesting article. This article is about developing and testing a novel self-reported questionnaires to evaluate household exposure to wood smoke in developed country. The reviewer 1 commented about the methods for the factor analysis and adding correlation analysis results to the manuscript. Hope author respond this issue in the revised manuscript.

Reviewers' comments:

Reviewer's Responses to Questions

**Comments to the Author**

1. Does this manuscript meet PLOS Global Public Health’s publication criteria? Is the manuscript technically sound, and do the data support the conclusions? The manuscript must describe methodologically and ethically rigorous research with conclusions that are appropriately drawn based on the data presented.

Reviewer #1: Yes

2. Has the statistical analysis been performed appropriately and rigorously?

Reviewer #1: Yes

3. Have the authors made all data underlying the findings in their manuscript fully available (please refer to the Data Availability Statement at the start of the manuscript PDF file)?

Reviewer #1: No

4. Is the manuscript presented in an intelligible fashion and written in standard English?

Reviewer #1: Yes

5. Review Comments to the Author

Reviewer #1: This paper develops and validates a new questionnaire of household exposure to wood smoke (WS). The sample is composed of 149 participants from the Western U.S. who use wood stoves for heating their household. Additionally, 30 of these participants had also their homes monitored for the presence of harmful pollutants in the air. Out of the initial 12-item scale, three items were discarded by the validation procedure, which included exploratory factor analysis and tests of associations between items. The paper finds that the scale is actually composed of three different factors (duration of, proximity to, and intensity of the wood smoke). Also, intensity was associated with respiratory symptoms and levoglucosan levels whereas duration was correlated with stove and flume maintenance.

This paper would make a fine addition to the literature. It is definitely helpful to have a validated scale to measure WS exposure. Also, the paper is well-written and easy to read. However, I have a few questions for the authors.

1. My main comment is about the possibility that measurement error may not be adequately captured by the factor analysis. Namely, I am curious about whether adding a methods factor to the measurement model in a CFA would reveal different associations between items and perhaps even support a two-factor solution where the methods factor is accompanied by a substantive factor (the HEWS scale). The sample size is definitely on the smaller end for CFA but I would like for the authors to explore the possibility that the different response options (“Never to Always” vs. the duration response options vs. the distance response option) are a possible source of measurement error that can be picked up and partialled out.

2. It would be interesting to add a table with the raw correlations between each individual item in addition to the correlation between each item and the rest of the items. This additional table could perhaps go in the appendix.

3. Data availability: Why is it not possible to make the replication data available to the journal? They could be appropriately deidentified so as to protect human subjects and comply with the standards of the journal and the academic community at large.

I would be happy to see a revised version of the manuscript, particularly if the anonymized data can be made available as per the journal guidelines.

6. PLOS authors have the option to publish the peer review history of their article (what does this mean?). If published, this will include your full peer review and any attached files.

**Do you want your identity to be public for this peer review?** For information about this choice, including consent withdrawal, please see our Privacy Policy.

Reviewer #1: No

---

## [Decision Letter · Decision Letter 1]

27 Dec 2022

Testing of a Novel Questionnaire of Household Exposure to Wood Smoke

PGPH-D-22-01097R1

Dear Professor Meek,

We are pleased to inform you that your manuscript 'Testing of a Novel Questionnaire of Household Exposure to Wood Smoke' has been provisionally accepted for publication in PLOS Global Public Health.

Best regards,

Changwoo Han, M.D., Ph.D.

Academic Editor

Reviewer Comments (if any, and for reference):

Reviewer's Responses to Questions

**Comments to the Author**

1. If the authors have adequately addressed your comments raised in a previous round of review and you feel that this manuscript is now acceptable for publication, you may indicate that here to bypass the “Comments to the Author” section, enter your conflict of interest statement in the “Confidential to Editor” section, and submit your "Accept" recommendation.

Reviewer #1: All comments have been addressed

2. Does this manuscript meet PLOS Global Public Health’s publication criteria? Is the manuscript technically sound, and do the data support the conclusions? The manuscript must describe methodologically and ethically rigorous research with conclusions that are appropriately drawn based on the data presented.

Reviewer #1: Yes

3. Has the statistical analysis been performed appropriately and rigorously?

Reviewer #1: Yes

4. Have the authors made all data underlying the findings in their manuscript fully available (please refer to the Data Availability Statement at the start of the manuscript PDF file)?

Reviewer #1: Yes

5. Is the manuscript presented in an intelligible fashion and written in standard English?

Reviewer #1: Yes

6. Review Comments to the Author

Reviewer #1: The authors adequately addressed my comments. Congratulations on a very nice paper.

7. PLOS authors have the option to publish the peer review history of their article (what does this mean?). If published, this will include your full peer review and any attached files.

**Do you want your identity to be public for this peer review?** For information about this choice, including consent withdrawal, please see our Privacy Policy.

Reviewer #1: No
